# Integration of Psychosocial Theory into Palliative Care: Implications for Care Planning and Early Palliative Care

**DOI:** 10.3390/cancers16020342

**Published:** 2024-01-13

**Authors:** Thomas V. Merluzzi, Natalia Salamanca-Balen, Errol J. Philip, John M. Salsman, Andrea Chirico

**Affiliations:** 1Department of Psychology, University of Notre Dame, Notre Dame, IN 46556, USA; nsalaman@nd.edu; 2School of Medicine, University of California, Los Angeles, CA 90095, USA; ephilip@mednet.ucla.edu; 3Department of Social Sciences and Health Policy, Atrium Health—Wake Forest Baptist Comprehensive Cancer Center, Wake Forest University School of Medicine, Winston-Salem, NC 27157, USA; jsalsman@wakehealth.edu; 4Department of Social and Developmental Psychology, Sapienza University of Rome, 00185 Roma, Italy; andrea.chirico@uniroma1.it

**Keywords:** palliative care, agency, relinquishing control, quality of life, coping, religion/spirituality, cancer, supportive care, hope, uncertainty

## Abstract

**Simple Summary:**

Palliative care focuses on relief from physical symptoms as well as from emotional, social and spiritual distress. This article promotes the importance of integrating psychosocial theories into standard palliative care to improve the quality of life of patients and carers. The psychosocial theories include the following: (1) *Agency*, which focuses on the empowerment of patients and carers to participate in palliative care and cope with the uncertainty of the outcome of the patient’s illness; (2) *Optimal Matching*, which focuses on the matching of the provision of supportive care with the needs of the patient and carer to improve the patient’s quality of life; and (3) *Hope*, which focuses on hoped-for outcomes throughout all aspects of palliative care. This theory-based integration may help patients, carers and healthcare providers to improve outcomes in terms of physical, emotional, social and spiritual quality of life as well as care planning.

**Abstract:**

Palliative care improves patients’ symptoms, quality of life and family satisfaction with caregiving, reduces hospital admissions and promotes alignment of medical care with the patient’s needs and goals. This article proposes the utility of integrating three psychosocial theories into standard palliative care with implications for care planning, early palliative care and optimizing quality of life. First, Control Theory focuses on the complex juxtaposition of promoting agency/empowerment in patients and carers and coping with often highly uncertain outcomes. Second, Optimal Matching Theory accounts for the alignment of need and provision of care to potentiate the quality of life effects of supportive care in a complex social process involving health care providers, patients and carers. Third, Hope Theory represents a dynamic process, which is marked by variation in the qualities of hope as the patient and carer confront challenges during palliative care. Future work will be translational in nature to adapt both assessment and interventions based on this theoretically driven augmentation of palliative care as well as to evaluate whether it provides a conceptual framework that has incremental utility in palliative care planning.

## 1. Introduction

Palliative care services for patients diagnosed with cancer, particularly outpatient palliative care, has increased in both National Cancer Institute (USA)-designated [1] and non-National Cancer Institute cancer centers. Generally speaking, the growth in palliative care has been fueled by research that has shown the substantial benefits of palliative care [2,3], and early referral to palliative care can augment these benefits [4,5,6]. Current evidence suggests that palliative care improves patients’ symptoms and quality of life, augments family satisfaction with caregiving, reduces excessive hospital admissions and costs and provides greater alignment of medical care with the patient’s needs and goals. Moreover, the focus on hospice care near the end of life has shifted to supportive care earlier in cancer illness trajectory. Thus, while palliative care in the past has been associated more closely with hospice care, that is no longer the case as palliative care is beginning to be provided earlier in the cancer trajectory, even as soon as diagnosis, and also as an adjunct to curative treatments [7].

Further, there is often a multidisciplinary team that provides concerted attention to patient and family needs to optimize the quality of life and reduce suffering that can be caused by significant medical, psychosocial and spiritual distress brought on by cancer [7]. The structure and organization of palliative care has been covered in comprehensive descriptive reviews of the types of palliative care delivery (e.g., outpatient, community based, inpatient consultation) and integrative models, which describe the integration of palliative care in the trajectory of cancer care [8]. Moreover, with an increase in the provision of early palliative care, the goals of palliative services may include adjuvant supportive care as an adjunct to active treatment regimens that have curative goals.

The goal of the current article is to build upon standard palliative care by integrating psychosocial theory and evidence that provide an augmented framework for the conceptualization and implementation of palliative care services. Thus, the focus of this work, in contrast to discussions of structural or integrative models, is on the role of psychological and social concepts in the context of the provision of palliative care, particularly early palliative care, with the goal of optimizing quality of life.

The addition of a psychosocial framework to palliative care, especially early palliative care, that is presented here is inclusive of the psychologist’s role in palliative care, which has been amply covered in other work on the provision of services [9,10,11,12] and shall not be covered here. That is, the current proposed integration of a psychosocial framework of palliative care is very amenable to the inclusion of psychological services, including for early palliative care; however, the main goal of the current paper is to provide a larger framework for palliative care that builds upon the integration of psychosocial theory and supportive evidence. While the presentation of the integrative model is intended to be useful both in terms of theory and the clinical practice of palliative care, it does not contain as yet a specific plan for implementation. However, after the discussion of the conceptual components of the model (Figure 1), a narrative and tabular (Table 1) exposition is presented to flesh out how the model might work.

### 1.1. Broadening the Scope of Supportive Care: Early Palliative Care

Recent efforts in oncology have sought to extend palliative care to a broader range of cancer care, including the promotion of early referral to palliative care. As noted prior, early palliative care improves patients’ symptoms and quality of life, augments family satisfaction with caregiving, reduces excessive hospital admissions and costs and provides greater alignment of medical care with the patient’s needs and goals [7]. Whereas typically the progression of care has focused primarily on the end-of-life, early referral to palliative care has the potential to improve clinical outcomes, including less acute care and aggressive care use and fewer acute hospitalizations [4], episodes that can be traumatic for both the patient and carers. Moreover, in states where access to palliative care is mandated, there is a much higher probability of dying at home or in hospice care surrounded by family, which is considered a quality outcome metric in public health research [13].

The benefits of early palliative care reflect a model that focuses on the patient experience by using their needs as the centerpiece of service provision as opposed to the sole use of prognostic non-curability [5] as a referral milestone. This approach assumes that the palliative model can exist side-by-side with curative goals, as well as in cases where the disease may be deemed incurable. This approach also aligns with the rapidly changing treatment landscape in oncology, most notably, the development of immunotherapies, which are revolutionizing the treatment of certain cancers once deemed untreatable or incurable. As targeted therapies (i.e., therapies that are designed to impact specific cancer cells) and immunotherapies (i.e., therapies that generally boost the immune system to target cancer cells) are approved for first line treatment, either in place of or in conjunction with chemotherapy, surgery and/or radiation, the trajectory of the disease is likely to be different from historical disease outcome trends. Similarly, outcomes may be further enhanced by biomarker profiles that determine higher probabilities of response to particular immunotherapy treatments in conjunction with established chemotherapy regimens. Essentially, while these new developments will bring hope for positive outcomes to many patients, they may also introduce more uncertainty in terms of expected outcomes regardless of the stage of the disease at diagnosis.

Despite the empirically based benefits of palliative care, especially early palliative care, barriers to access continue to exist, including a lack of clinical resources, health care providers’ reluctance to refer, a lack of training in palliative care [14], patient and carers’ reluctance to accept referral, restrictions on eligibility and long-standing racial and ethnic health care disparities. Despite these challenges, there is little doubt that palliative care will continue to grow, and its utility will become an integral aspect of cancer care. With this growth, there will be a need to examine the impact of palliative care on patients and carers, particularly in the context of the increased implementation of early palliative care where prognostication of disease outcomes is more indeterminant, and therefore, uncertainty of outcomes becomes a more salient aspect of care.

### 1.2. Shift from Prognostic to Needs-Based Determination for Referral to Palliative Care

As noted above, in the not-too-distant future, palliative care may be provided in a context of increasing uncertainty [15,16] regarding the trajectory of illness. In this regard, there is growing discussion regarding the point at which palliative care should be integrated into cancer care and how decisions are made to initiate palliative care. Such decisions will be more fraught with the increase in the availability of early palliative care and as new treatments continue to be discovered and refined. The discussion regarding referral decisions revolves around three potential models [5]: (1) prognosis-based, (2) needs-based and (3) a hybrid of the prognosis and needs-based approaches. As palliative care is implemented as an additional layer of supportive care, as advocated by the Center to Advance Palliative Care, the needs-based model is more appropriate than the prognosis-based model, which has been restricted to a time frame and may overlook patients’ actual needs and the rapid advances in oncologic treatment modalities. The needs-based approach is consistent with the World Health Organization’s perspective on palliative care, which advocates for the prevention and relief of physical, psychological, social and spiritual suffering. If a needs-based approach is adopted and early palliative care is increasingly utilized, cancer care will be imbued with an inherent juxtaposition of patient care that emphasizes patient agency/activation and uncertainty regarding disease outcomes [17].

### 1.3. Agency and Uncertainty

In juxtaposition to the aforementioned uncertainty of the illness trajectory in early palliative care, there are new models of care [18] that focus on patients not as passive recipients of care in an environment of uncertainty but as agents of their health and collaborators with those who provide care [19,20]. This perspective is based on western philosophical assumptions of autonomy (e.g., Immanuel Kant and John Stuart Mill; [21]), which espouse that essentially, within some limitations, every person lives life in accordance with reasoning, motives and wishes without undue manipulation from external forces. Moreover, inherent to this perspective is the assumption that the patient role is marked by agency, empowerment and activation [22], and a further implicit assumption is that this role is universally important to every patient. These empowerment models and assumptions give rise to some issues for patients and their carers.

First, cancer is a disease that often imposes some limits on autonomy and agency with regard to one’s health and lifestyle [23]. Therefore, there may need to be a discernment process by which a patient (and perhaps carers) defines the parameters of empowerment and agency in broad terms as well as in specific behaviors in everyday life. Moreover, this discernment process includes deciding what to control and what may not be worth controlling perhaps because controlling certain things is too difficult, not possible, or too far into the uncertain, distant future [20]. This discernment process may also be based on a distinction between symptom control, which may be perceived as more controllable (e.g., taking medication, participating in physical therapy) versus disease management, which may be perceived as a long-term undertaking with less certain outcomes. Thus, there may be a dynamic relationship between taking control of some things and letting go of control of others [24], which may change with the trajectory of the disease. However, this sense of control may be very different for those who have more resources than those who have been typically marginalized with regard to health care, particularly in situations where access to health care and medical mistrust may be barriers to engaging in community-based, home health care.

Second, in the context of cancer, control also typically has relational components, where the patient is embedded in a social network (i.e., family, carers, health care professionals) such that agency and empowerment are both personally and socially constructed [25], negotiated and revised. Thus, the taking and letting go of control may also interface with social support such that there is a balance between need and provision of social support [26], as well as the process of taking and letting go of control for the patient, carers and palliative care team members.

Thus side-by-side, there exists the process of assuming agency in one’s own care and the uncertainty of outcomes with the potential for a disconnect between one’s personal agency and the feasibility of hoped-for outcomes. Managing this juxtaposition between personal agency and future outcomes is a critical component in the integration of psychosocial theory into palliative care, which includes the impact of a support system that plays an important mediating role in that dynamic of personal agency and outcomes.

## 2. Integrating Psychosocial Theory into Palliative Care

### 2.1. Overview

When considering the totality of palliative care, and especially early palliative care, there are three components that form the proposed integration of psychosocial theory into palliative care (Figure 1). First, there is a paradox involving control and uncertainty that would benefit from conceptualization that might help untangle the apparent dilemma of being encouraged to be in control in a context where uncertainty looms large. Thus, patients as agents of their care, as opposed to mere recipients of care, can be helped by a conceptualization in which the juxtaposition of agency/control and uncertainty is explicitly noted and utilized to promote disease management and improve psychological, social and spiritual quality of life.

Second, as noted earlier, at a deeper level of the provision of palliative care is the fact that most patients are part of a social network that may impact to what extent patient agency can be adopted [19] even if it is valued by the patient. Thus, autonomy and control in the context of palliative care necessarily have relational components that may affect agency—even more so in the context of serious illness where patients may experience physical limitations and where doctors and medical care generally may decrease control and autonomy [27]. Also, autonomy has been traditionally gendered, which makes for different narratives based on gender regarding the meaning and exercise of autonomy [27]. In summary, the coordination of the provision of care such that the matching of needs and provision of support are optimized, represents a critical component of palliative care. Moreover, this dynamic process takes place in a social network that involves the patient, carer and palliative care support team.

Finally, hope is a pervasive concept in that it provides a platform for perceiving the future, taking into account the uncertainty of outcomes. However, hope is not a uniform experience; it is nuanced and qualified by the context in which hope is expressed. A new integrative theory of hope [28] that is based on the dimensions of control and uncertainty presents the confluence of those dimensions as the basis for defining qualities of hope that may be incorporated into palliative care, including early palliative care.

The integration of psychosocial theory into palliative care combines philosophical perspectives and modern psychological control theory as well as the social dynamics of caregiving. This framework also provides a role for hope in the process of providing palliative care. The result is an integrative model that adds three critical components to the standard model of palliative care: (1) the juxtaposition of control and uncertainty, especially in early palliative care; (2) the dynamics of matching need and provision of support; and (3) the qualities of hope in the context of palliative care (Figure 1). This integrative model provides a promising framework to help guide the planning and implementation of palliative care, including early palliative care interventions, with the goal of optimizing the quality of life. A presentation of each component theory is followed by a strategic integration for the purposes of framing palliative care, including early palliative care. The model also includes three phases (i.e., initial, mid-phase and advanced) of palliative care that are integrated with the three theoretical components.

### 2.2. Component 1: Modern Control Theory: Perspectives on Personal Agency and Uncertainty in Palliative Care

Inherent in palliative care are two types of control: (1) primary control, which is tantamount to the exercise of personal agency in order to change the circumstances that result in certain outcomes, and (2) secondary control, which focuses on accepting one’s situation or externalizing the control of outcomes. Primary control describes the new model of palliative care, which empowers patients as much as possible to exercise personal agency and cope actively (e.g., problem-solving coping) and change their circumstances, whereas secondary control fosters acceptance and meaning (e.g., emotion focused-coping), which may reduce distress but does not change the circumstances. However, secondary control also includes placing control in the hands of powerful others (e.g., physician, God, universe, science). Thus, trust in physicians might mean viewing longer-term outcomes as under the control of others’ expertise or science.

There are also spiritual [29] and secular forms of secondary control [30]. In the former, one may construe the universe as having order in the form of cause and effect [31], thus trusting that that order will account for longer-term outcomes such as longevity, and the latter is based on practices, such as mindfulness, which reduce emotional reactivity [30] by focusing on the present rather than the future, thus eschewing worry about longer-term outcomes. In the context of religious coping, placing outcomes in “God’s hands” is tantamount to “letting go” of personal responsibility for outcomes based on the belief that God may influence outcomes. There is emerging evidence to support the utility of this form of secondary control that fosters quality of life [24,32].

In line with new models of palliative care, primary control is useful in engaging patients in collaborating and constructing their well-being within the parameters of the illness, which may be regressing, resolving or uncertain, by managing current symptoms that may interfere with treatments and quality of life. At the same time, secondary control may provide a means to cope with the uncertainty of longer-term outcomes such as the disease trajectory or longevity. This approach may provide solace in that outcomes are no longer as worrisome or the responsibility of the patient. Thus, assuming personal control and collaboration regarding treatment regimens to manage symptoms and enhance quality of life would focus on the patient’s current care. Secondary control, including relinquishing or suspending control of outcomes, would be invoked to manage feelings of anxiety and worry about the longer-term outcomes, which are fueled by uncertainty.

Relinquishing or suspending control of future, hoped-for outcomes has a long history from ancient China, medieval Christianity and Stoic philosophy to modern theology and psychology as well as across a number of religious traditions and in the secular realm. The Stoics, Seneca, Epictetus, and Marcus Aurelius clearly distinguished our responsibility to be fully accountable for our own behavior from what they believed to be a fatalistic world where outcomes in life were a function of the “gods” or fate. Thus, in the context of uncertainty or a lack of controllability of outcomes, we are required to “do good and act in conformity with reason” ([33] p. 127). In the worldview of the Stoics, probity is required and then one must let go of the outcomes. This process resembles the processes of many people with cancer who cope by participating in medical and complementary medicine practices, including spirituality, hoping that the cumulative effects have some bearing on disease outcomes but also being keenly aware of the uncertainty inherent in this process. Along those lines, religious and spiritual coping is often exemplified by adherence to medical regimens but also detaching from outcomes by vesting them in universal forces or God. Clearly, there is a similarity of this worldview to that of the Stoics. Furthermore, mindfulness may be the contemporary extension of letting go [34] in interventions for serious health problems, and some forms of contemporary therapy such as Acceptance and Commitment Therapy embody a realistic engagement in life by not being overwhelmed by negative emotions and challenging situations [35]. Thus, “letting go” of long-term outcomes has benefits in terms of quality of life [24,32].

The implications for palliative care are increasingly evident. As need-based (versus prognosis-based) palliative care becomes a more integral part of the referral process, especially referral shortly after diagnosis, outcomes will be more uncertain [36]. Health professionals can introduce a Stoic-type perspective in which patients are encouraged and supported in adherence to the regimens of care and at the same time encouraged to let go of or suspend control of longer-term outcomes. At the beginning of this process, in addition to medical treatment regimens, the care plan can be centered on immediate need-based palliative care for symptom control and solving immediate problems, whereas disease outcomes (e.g., being cured), especially with the expansion of immunotherapies, may remain relatively uncertain. The emerging research on “letting go” or relinquishing control of outcomes versus assuming responsibility for outcomes seems to point to a number of benefits: better current quality of life, less depression, viewing life as more meaningful and better coping [24,32]. This conceptual framework based on philosophical tradition, modern psychological control theory and religious or spiritual maxims provides an accessible path to navigate through a very difficult time both in terms of exercising agency in managing immediate care needs and reducing distress about uncertain future outcomes. Clearly, this path will be fraught with challenges and difficulties that will require some revisions in care to optimize immediate outcomes, but the overall model gives health care providers, carers and patients a framework for how to traverse this difficult phase of their lives while, at the same time, reduce worry and anxiety about the longer-term outcome of their illness.

In summary, the initial stages of palliative care may present a dilemma of encouragement to exercise agency and tolerate an uncertain future. Control theory provides a framework for considering the juxtaposition of these two forces—one that can help promote engagement in medical regimens and care to improve physical and emotional quality of life and at the same time help promote disease adjustment and accept a future with inherent uncertainties [37,38,39]. It is important to mention that in the US, in particular, secondary control, placing control with others (i.e., health care providers), may present an interesting conundrum for marginalized populations who may have an understandably low amount of trust in medical institutions/systems. However, religious forms of secondary control (i.e., placing outcomes in “God’s hands”) may be a familiar approach to coping. In the context of community-based palliative care, which takes place mostly in the homes of those receiving care, the onus for recognizing mistrust should be not on the patients themselves but on the palliative care team to create a culture of care that is expert but also kind and trustworthy.

### 2.3. Component 2: Optimal Matching of Need and Provision of Supportive Care

As noted in the discussion of the first component, agency/uncertainty, there is a complex arrangement in which the patient and carers are expected to take an increasing amount of responsibility for care. First, that transfer of responsibility to the family may not be a model that carers wish to adopt fully and, moreover, even if it is adopted, there are family dynamics that may result in the patient and carers being in concert or conflict over their roles in this complex social system of patient, family carers and health professionals. For example, uniform deference to the carer, especially in cases where the patient and carers’ goals are in conflict can undermine and decrease patient empowerment and quality of life.

In these multi-level dynamics, health professionals’ relationships with the carer and patient and also carers’ relationship with the patient play a critical role in the provision of services and personal care and presumably the optimization of the quality-of-life outcomes for the patient and family. This delicate balance may be cast into an Optimal Matching Model [40] of support where the need for supportive care and its provision are optimally matched [26], better quality-of-life outcomes are facilitated than when they are mismatched. Thus, in the context of palliative care, there are two levels to optimal matching of supportive care. On one level, there is the optimal matching of supportive care provided by health care professionals with the needs of the carer and the patient. On the second level, there is the optimal matching of support provided by the carer with the needs of the patient.

In the Optimum Matching Model, the excessive provision of support (either by professionals to the patient or carer OR by the carer to the patient) when it is not needed, may reinforce a passive patient role or a helpless carer role and undermine agency as well as contribute to deterioration of functioning on the part of the patient or carer. Alternatively, the failure to provide support to the patient or to the carer when it is needed is also problematic. Not only does this optimal matching need to be assessed and constantly monitored, but also the patient and carer can be empowered to engage in the assessment process to foster optimal matching of the need and provision of support [25] to facilitate the desired quality of life outcomes. Thus, optimal matching would be tailored to each patient and his or her carers, taking into account their customary patterns of decision making as well as goals regarding facilitating agency in both the patient and carer.

Optimal Matching Theory may not be common knowledge among palliative care providers; however, there is a recognition of the issues of roles and responsibilities that has been a focus of discussion [41]. These issues of coordination of roles and the communication needed to consistently clarify roles have focused on the professional side of palliative care, but the same arguments may be made with regard to the patient and carers. That is, there may be some confusion based on the informality of roles in the home setting but also conflicting understandings of family members’ needs by the palliative care team, particularly that of not being overburdened.

Thus, for optimal matching of the need and provision of supportive care, the care team would have to meticulously assess the needs of the carer and patient in concert with the carer and patient, then propose a two-level plan. On one level, there is the implementation of a formal written care plan which describes the needs of the carer and patient and a written consensus-driven provision plan among the professional staff that is then shared and revised with input from the carer and patient. The next level of optimal matching of the need and provision of support would require the palliative care team to facilitate the same discussion between the carer and the patient. Clearly, for some carers, the collaborative model may not fit with their traditional notions of support in which the carer defines support in a way that may undermine the capabilities and autonomy of the patient, thereby reinforcing a passive patient role. Alternatively, a carer may not want to assume some responsibilities and the patient may not want the carer to have to perform some procedures that invade privacy or violate their sense of decorum.

In summary, optimal matching at both levels (professional with carer/patient and carer with patient) is difficult work that requires education, communication and consensus among the professional team as well as facilitation of the optimal matching between the carer and the patient. Optimal matching is also a dynamic iterative process that is constantly changing as conditions change in the lives of the carers and patients [42,43]. These transitions would call for re-evaluation and adjustments to the optimizing model, which would be expected to become part of the process over time. Finally, this dynamic process of matching need and provision of supportive care involves a complex social network of communication between the patient, carer and palliative care team [44,45]. While the description provided in this article is more theoretical than applied, there are ample resources on specific communication processes in the context of palliative care that may be used to achieve the goals of this component, namely, the matching of the need and provision of supportive care to optimize quality of life [44,45].

### 2.4. Component 3: Dimensions of Hope in the Context of Palliative Care

In addition to the agency/uncertainty issue and optimal matching of the need and provision of care, an understanding of the dynamics of hope can also play a critical role in palliative care, especially in early palliative care. Hope is an essential component of palliative care in that hope has been related to less distress and fewer physical symptoms [46] and is positively associated with resilience, effective coping and quality of life in those with serious illnesses [47]. In spite of these positive qualities of hope, theories of hope have been overly restrictive in purview [48] or overly general and, as a result, do not provide a comprehensive or contextualized picture of hope [49]. In contrast, the development of a new integrated theory of hope [28] based on established conceptual principles (control and uncertainty) and applicable to a variety of circumstances could provide guidance to palliative care providers, patients and carers, especially in the context of early palliative care. Indeed, this new integrated theory of hope includes an appraisal process that assesses primary and secondary control and possesses useful descriptions of qualities of hope that are based on the intersection of control and uncertainty. Thus, this new hope theory is highly integral to a control theory approach to current models of palliative care and is therefore integral to the first theoretical component, which focuses on agency and uncertainty.

## 3. Overview of the New Integrative Theory of Hope

On the general level, the proposed hope theory [28] is based on the assumption that hope refers to a future good that may, at times, be difficult but possible to obtain. Moreover, that future good (or desired goal) must be specific and meaningful to the individual. At a more micro-level of the theory, uncertainty and control are key drivers of hope and integrate relevant psychological processes, including appraisal, coping and meaning. Thus, hope is a relatively confident expectation about a positive outcome that varies in uncertainty, while also varying in the relationship of personal control to that outcome. The process of hoping includes an appraisal of the utility of primary and secondary control and, therefore, is related to the first component which involves agency and “letting go.” Primary control is mainly related to agency/self-efficacy, which enhances the probability of the hoped-for outcome. Alternatively, secondary control involves vesting the hoped-for outcomes in others (physician, science, God) when personal efforts to achieve the outcome may not be sufficient. In situations with high uncertainty and low personal control, importance is placed on secondary control strategies such as transcendence with regard to the hoped-for outcomes, whereas in situations with low uncertainty and high personal control, importance is placed on personal agency/self-efficacy in realizing the hoped-for outcomes.

An examination of Table 2 reveals qualities of hope that are a function of certain combinations of personal control and uncertainty and have utility in the context of palliative care. While Table 2 contains four quadrants, the model is actually a circumplex; however, for descriptive purposes, the four quadrants provide a more accessible interpretation of the intersection of control and uncertainty. Also, it is not intended as a purely stage model of hope in that one might experience more than one quality of hope at any one time and experience them in no particular order or in a step-wise fashion. However, it is useful to express hope from the perspective of the desired outcome, in which case, four specific control–uncertainty scenarios may be described accordingly (see Table 2).

First, in high control and low uncertainty situations (Q1), the individual’s active behaviors are effective in achieving the desired outcome; hence, self-efficacy/agency is the driving factor in the hope process, and it is tantamount to primary control, where one’s behavior is closely tied to the outcome. In this case, hope is vested in self-efficacy—the competent execution of behaviors that lead to a hoped-for outcome. Second, in situations where there is high control and high uncertainty (Q2), achieving the desired outcome becomes more challenging than in Q1 because those situations may require more persistent active efforts to achieve the outcome. In this case, hope is vested in perseverance, endurance, flexibility and resilience, which are consistent with primary control but over the longer time frame that it takes to achieve the outcome. Third, in situations of high uncertainty and low control (Q3), the individual realizes that achieving the hoped-for outcome does not depend solely on personal effort, and hope may depend on secondary control strategies such as finding meaning, placing outcomes in the control of others (e.g., doctors and medical science,) or by adopting a perspective that someone or something “greater” than one’s self has control. Thus, in this situation, hope is vested in external powers, which might include transcendent forces such as the universe or God. Finally, there are situations in which control is low and uncertainty is also low (Q4), where the individual may disengage from effort, accept what seems to be an inevitable outcome and/or perhaps finds meaning or benefit in that outcome.

As noted earlier, these qualities of hope are not intended to form some linear path or steps in the hope process; although, there may be some progression in certain cases. In the context of cancer care, at any one time, patients and carers may engage in various hope processes simultaneously that reflect, for example, hope for completing manageable tasks, such as draining fluids using a Pleurx kit (Q1); hope for outcomes based on endurance (e.g., the ability to persist in tolerating a treatment over a long period of time in spite of persistent negative side effects) (Q2); hope based on transcendence such as placing hoped-for outcomes, (e.g., remission, longevity) in “God’s hands” or in a new experimental treatment regimen (Q3); and hope for solace in accepting an outcome (e.g., that there is no cure) and coming to terms with that reality (Q4). In all of these examples, there is a confluence of personal control and uncertainty that informs qualities of hope. Moreover, an understanding of these qualities of hope on the part of health care providers, carers and patients can at times provide a rationale and narrative for encouraging a patient and carer to “hang in there” through some difficult times when managing pain or fatigue (Q2: hope vested in endurance) and also by encouraging patients and carers to not focus all their attention on outcomes, by suspending, relinquishing or referring control of longer-term outcomes to others (Q3) as a way of reducing distress [50]. This re-evaluation process has been described as “re-goaling” where the patients and carers transition from one set of goals to another, in a process of maintaining hope [51].

## 4. Integration of the Components: A Hypothetical Narrative with Implications for Psycho-Social Processes in Palliative Care

### 4.1. Integration of Psychosocial Components with Phases of Palliative Care

An overview of the integrative model is presented in Table 1, which contains the three psychosocial components (agency/uncertainty, optimal matching of need and support and hope) and three phases (initial, mid-phase and advanced). While the three psychosocial components have been described in some detail, they are theorized to occur in the four dimensions of palliative care (spiritual, physical, psychological and social) (see Figure 1). Moreover, the assumption is that the model is initiated with the start of the provision of early palliative care. However, a late referral to palliative care can result in the model starting at mid- or even advanced phases, which is more commonly the case. Importantly, while the phases may be related to the trajectory of cancer, they are not identical concepts. The phases refer more to the kind of care that is needed as opposed to the trajectory of the disease. Thus, for the same disease stage, one person may be provided care consistent with the initial phase (Table 2), whereas another might need supportive care more consistent with the mid-phase of the model.

In the initial phase, the goal of the first component (agency/uncertainty) is to focus *on patient and carer agency* and optimize shorter-term outcomes typically related to physical symptom management. Thus, there might be a less specific discussion at this point about prognostic outcomes compared to later phases. That is, for the most part, the palliative care team members present the importance of patient and carer agency in the management of symptoms in routine care and assess the patient’s and carer’s understanding and willingness to “buy into” that model of care, whether the patient is receiving services at home, as an outpatient or an inpatient. The juxtaposition of encouraging agency/empowerment and the uncertainty of outcomes may rise to the level of a dilemma at this point, but the focus remains mostly on empowering the patient and the carer to implement a care plan that involves taking on many tasks to improve the physical quality of life of the patient while also not burdening the carer with stress.

Moving to the second component in the initial phase (Optimal Matching: Patient–Provider–Carer), a thorough *assessment of the needs* of the patient and the patient’s capability to participate in meeting those needs, as well as the capability of the carer to provide support for those needs, sets the stage for the optimal matching of the need and the provision of care in order to optimize quality of life outcomes. As noted earlier, this component interfaces with the first component in the sense that patient and carer agency set the stage for optimal provision of care. Too much support, when it is not needed, can undermine the agency of the patient and foster a helpless patient role. Too little care, when it is actually needed, may jeopardize the well-being of the patient. This matching of need and provision also involves the palliative care team in that the needs of the patient and carer and the provision of professional care operate under the same premise; that is, not all support is optimally helpful, and the positive impact of supportive care is greatest when the need and provision are matched. Thus, at this initial level, the goal is to foster and support agency in the patient and the carer, while at the same time, developing a plan that optimizes the matching of need and provision of care to realize primarily physical quality of life outcomes. Hope at this initial phase is vested in the personal agency (Table 2, Q1) of the patient and carer to carry out the tasks of the need-and-provision plan.

The mid-phase of the model may be triggered by a number of events or thoughts that might include frustration with progress, “personal agency” fatigue where patients or carers may be weary from the efforts to carry out the care plan, worries about future outcomes, which may include questioning the uncertainty of disease outcomes, and needing some respite from the constancy of care and for self-care [52]. The patient and carer may be questioning their endurance in the process of maintaining personal agency and maintaining hope in the face of negative thoughts, emotions and physical symptoms. Even without articulating the issue, the patient and carer may be questioning the relationship between the endurance it takes to stay focused on the short-term goals of symptom management and future outcomes. As opposed to the first phase, where digging into the tasks seems reasonable, particularly without much reference to long-term disease outcomes, the mid-phase gives rise to questioning personal efficacy as uncertainty looms, and the work of taking care of one’s self and the provision of care become more difficult to sustain due to perhaps physical, emotional and/or spiritual distress. In this mid-phase, there may be a need to normalize and affirm the debilitating effects of “personal agency” fatigue, emotional distress, and inadequacy of support and, at the same time, shore up the support and resources needed to maintain patient and carer well-being.

In this mid-phase, some aspects of hope may transform from a strong relationship between personal efficacy and outcomes in the initial phase to hope that is vested in endurance (Table 2, Q2). That is, the tasks are still very doable by the patient or carer, but the question of their relationship to outcomes becomes more salient as the uncertainty of the outcomes becomes more apparent. Support for resilience, tenacity and endurance may call for the involvement of more resources in order to keep the patient and carer engaged in the goals of care and to deal with the distress that may accompany personal agency fatigue and the diminution of hope. Thus, in the mid-phase, hope may transform into hope for endurance, tenacity and resilience to remain effective in engaging in the tasks of palliative care.

At the advanced phase, there may be an increase in worry or anxiety about disease outcomes as more questions arise regarding the goals of palliative care. These concerns may be a natural progression from the mid-phase where hope was vested in endurance, that is staying with the program, to the advanced phase where there may be a modification of the goals [51]. This transition may be accompanied by more conscious awareness of the “disconnect” between personal agency and outcomes, even if personal agency and carer agency with regard to treatment goals are improving symptoms and, therefore, physical quality of life. However, increases in weariness, worry and existential anxiety may cause both the patient and carer to over- or under-function with regard to self-care or the provision of care even if there is no imminent danger of death. This dysregulation may call for a re-examination of the need and provision care plan.

With regard to the worry about disease and longevity outcomes, there is mounting evidence that for those who engage in religious coping, turning outcomes over to God, as opposed to taking responsibility for outcomes, may imbue this advanced phase with meaning and peace that promotes physical quality of life [24,32]. Perhaps those whose belief system supports placing outcomes “in God’s hands” achieve some solace, because they believe that in that process, there is continuing effort on God’s part, which releases them from worrying about outcomes. While a secular version of this process has not been thoroughly tested, placing trust in medical science, that is relinquishing control of outcomes to others, may foster hope that may provide some peace. Moreover, the goal of mindfulness practices is to maintain a focus on the present versus an uncertain future, which may reduce worry about outcomes. Clearly, the religious version of relinquishing control of outcomes has value for people who hold those beliefs, but other forms of relinquishing control of outcomes may have utility. In these instances, particularly when outcomes are highly uncertain, patient and carers are asked to some extent, to let go of outcomes but persist in medical regimens and personal care that have as their goals optimizing physical and emotional quality of life. Hope in this situation (Table 2, Q3) is vested in transcendence (e.g., relinquishing control of outcomes to God or the universe) or in others (e.g., trust in medical science, physicians). Also in the advanced phase, there may be a sense of some diminution of personal control, which may foster a strong sense that there might be less actual connection between personal agency and outcomes. Finally, in cases where palliative care is an adjunct to active treatment, there may be some point at which the patient exits palliative care if that treatment regimen is progressing toward cancer remission and survivorship. Alternatively, if the trajectory is toward hospice care in the advanced phase, hope may be vested in acceptance (Table 2, Q4).

### 4.2. Integration of Psychosocial Components with the Dimensions of Palliative Care

In addition to integrating psychosocial theory into the phases (early, mid-phase advanced) of palliative care, the goal was to utilize psychosocial theory to conceptualize palliative care across the dimensions of physical, psychological, social and spiritual well-being (Figure 1) and, in doing so, enhance the provision of care. For example, in the standard practice of palliative care, there may be some medical regimens that are taught to a carer and patient to reduce problematic symptoms and enhance the physical well-being of the patient—such as using a Pleurx drain kit to reduce the fluid in the pleural cavity. This regimen can be cast in the context of the theoretical perspectives as well as the dimensions of palliative care (Figure 1). Thus, besides the physical comfort provided (relief of symptoms, less pain, easier breathing), the process of successfully draining fluid has a psychological component (e.g., the patient’s and carer’s agency for mastering the skills needed to perform the procedure, dealing with the stress and anxiety of performing a multi-step serile procedure, worry and uncertainty about its effectiveness beyond immediate relief, matching technical and social support with the level of agency of the carer and patient, hope that the procedure, when well performed, will be effective each time in relieving discomfort), a social component (the interaction and collaboration between the care staff, carer and patient to instill agency and confidence in performing the procedure, the interaction and mutual support of the patient and carer in managing the many intricate steps in the successful execution of the sterile procedure, the bi-directional dynamic matching of the provision of support to perform and endure the procedure on a daily basis over a period of time, hope that one can maintain the support needed to perform the procedure and endure the discomfort during the task) and a spiritual component (coping with uncertainty of outcomes, affirmation, social presence, connectedness, mutuality, hope in the form of faith and trust [53]). In summary, the integration of the three theoretical components into the phases and dimensions of palliative care may enhance not only the conceptualization of care planning but also care provision.

## 5. Future Directions and the Challenge of Health Disparities

### 5.1. Future Directions

The goal of the integration of psychosocial theory into palliative care, including early palliative care, is optimizing physical, emotional, social and spiritual quality of life. The theoretical structure includes the integration of some overarching concepts that are inherent in new models of palliative care: (1) the juxtaposition of personal agency with the uncertainty of outcomes, (2) optimizing supportive care with the matching of the need and provision of support that occurs in a complex, dynamic social environment that involves three parties, the patient, carer and health-care provider, and (3) qualities of hope that may transform with changes in agency and uncertainty and with the trajectory of cancer (Figure 1). This conceptual structure captures the complexity of palliative care, integrates well with early palliative care and, at the same time, has practical utility for the provision of care. After some refinement of this model, the next steps would be to develop an assessment battery to provide a profile of the patient and carer with respect to the three components and three phases of the model to aid in the implementation of palliative care that optimizes physical, psychological, social and spiritual quality of life outcomes. There are emerging assessment approaches that are specific to palliative care with respect to quality of life [54] and other aspects of palliative care [55] that could be adopted for use with this psychosocial integrative model. However, more work would be needed to integrate self-report measures with regard to satisfaction with the process of optimal matching of the need and provision of support, both from the patient’s and carer’s perspective.

Tailoring interventions to the patient and carer profiles could result in some prototypical interventions that could be developed for palliative care health professionals, patients and carers. Thus, the proposed integrative model would require program development and testing and also would involve more staff time for implementation than is currently the case. Those challenges may not be supported currently in the staffing and compensation for palliative care.

### 5.2. Health Disparities in Palliative Care

Whereas the use of palliative care has increased broadly in the US from 2004 to 2020; there remains a disparity in the provision of palliative care for ethnic-minorities that follows the same pattern of other forms of health care. For example, a recent study [56] showed that palliative care use by patients increased significantly over time, from 14.9% in 2004 to 27.6% in 2020. And, although increases were observed across all racial and ethnic groups, “non-Hispanic Black, Asian or Pacific Islander, and Hispanic patients were 13%, 26%, and 35% less likely to receive palliative care, respectively, than non-Hispanic white patients after adjusting for clinical and sociodemographic factors. There was no significant difference in palliative care use between non-Hispanic white patients and patients who identified as American Indian, Alaska Native, or other” [56]. Those disparities in the provision of palliative care disproportionally affect marginalized groups, placing them at risk for poorer outcomes. Moreover, under-resourced patients are often treated in community settings, therefore these groups (racial or ethnic minority patients, rural patients, etc.) would stand to benefit the most if the proposed model can equitably enhance their care as well. Clearly, the advantage of community-based palliative programs is their relative cost compared to institutionally based programs. Moreover, given that many people eligible for palliative care will increasingly be of Medicare-age, palliative care services apart from hospice care hopefully will be more broadly available. However, like all aspects of health care, closing disparities in the provision of palliative care and especially early palliative care is a critical aspect of improving the quality of life of all those with serious illnesses.

## 6. Conclusions

In this preliminary description of the integration of psychosocial theory, the focus is on optimizing the quality of life of patients and carers in the provision of palliative care, including early palliative care. The integrative model is intended to provide a conceptual framework for palliative care that includes a psychosocial perspective, which can provide the palliative care team as well as the patient and carer with concepts and procedures to optimize quality of life. While there has been a great deal written about planning, developing and integrating palliative care programs, the current integrative model is intended to infuse psychosocial, process-oriented concepts into the provision of palliative care. However, because this is the initial presentation of an integrative model, adapted assessment measures, manualized interventions and accessible media materials would need to be developed in subsequent steps in the translation of the model into the implementation of care planning and provision. Thus, this presentation represents the first steps in building the conceptual infrastructure of an integrated psychosocial process model of palliative care designed to improve the quality of life in patients and carers.

## Figures and Tables

**Figure 1 cancers-16-00342-f001:**
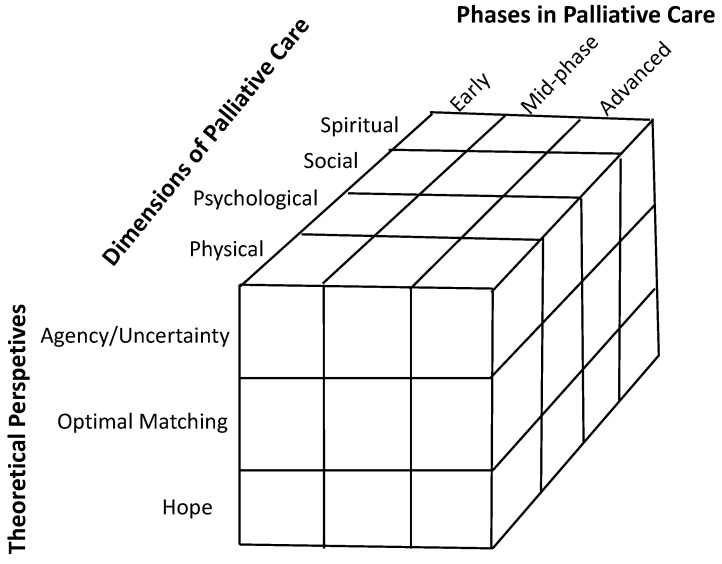
Integration of psychosocial theory (Agency/Uncertainty, Optimal Matching, Hope) into the standard palliative care dimensions (physical, psychological, social and spiritual well-being) and phases (early, mid- and advanced phases).

**Table 1 cancers-16-00342-t001:** Focal points in the integration of psychosocial theory into the provision of palliative care.

Component	Initial Phase and Goals	Mid-Level Phase and Goals	Advanced Phase and Goals
Agency—Uncertainty	- Foster the importance of patient agency and the support for that process from providers and carers. - Foster the importance of carer agency in the provision of support for patient’s agency. - Focus on provider’s role in supporting the carer’s agency. - Goal: Optimize, for the most part, short-term physical and emotional QOL outcomes by supporting patient and carer agency. Because of uncertainty of disease outcomes, there is less focus on long-terms outcomes.	- Foster acceptance of seemingly paradoxical processes: accepting what appears to be a division between personal agency, which involves patients’ assuming some control their care, and the uncertainty of longer-term outcomes (e.g., remission, longevity). - Provide support/affirmation for the unease this dilemma may foster and support shorter-term QOL of life goals (symptom management, emotional well-being). - May also be some “agency” fatigue if positive changes in well-being are not materializing or not quickly enough. - Goal: continue support for short-term goals and introduce tolerance of uncertainty for long-term outcomes.	- May be some “agency” fatigue if positive changes in well-being are not materializing. - Support resolving worry and anxiety about the disconnect of agency and uncertainty, concerns about effort and outcomes. - Strategies include “letting go” of outcomes. - Coming to terms with outcomes may be based on spirituality (e.g., meaning making, dignity, legacy), religion (e.g., deferring outcomes to God) or a mindfulness model, which fosters a focus on acceptance and the “here- and-now” versus the uncertain future. - Goal: emotional QOL, reducing distress and worry about uncertainty.
Optimal Matching: Patient–Provider–Carer	- Accurate assessment of the needs and capacity of the patient and the carer in order to match the provision of support with patient needs. - Revisit patient agency and needs as well as carer agency in matching need for and provision of support. - Goal: work toward match of need and provision to optimize short-term QOL of life outcomes.	- Focus on revising need and provision of support based on the reassessment of symptoms and modifications in care to optimize QOL outcomes. - Revise patient agency and needs as well as carer agency in matching provision of supportive care. - May also intensify the palliative care providers’ role in the provision of care. - Goal: work toward improving the match of need and provision of support to optimize short-term QOL outcomes.	- Focus on patient and carer roles in the optimal matching of need and provision of care: maybe increases in patient and carer weariness, worry, despair? - Carer over or under functioning? Increasing need for provider visits/care? - Goal: revise need and provision of care and provide particular attention to the patient’s and carer’s emotional QOL.
Hope	- Focus on agency (Q1: Section 3): what the patient and/or carer can do to foster short-term QOL outcomes (typically symptom management), - Hope is vested in agency: self-efficacy to optimize shorter term QOL outcomes.	- Focus on endurance (Q2: Section 3): support for tenacity or perhaps modification of the need/provision plan to maintain patient and carer hope for endurance and resilience in course of treatment regimens in spite of challenges. - Hope vested in endurance in care and treatment regimens in spite of difficulties, concerns and uncertainty of outcomes.	- Focus on transcendence (Q3: Section 3) or acceptance (Q4: Section 3). Engaging religious coping regarding the “letting go” of long-term outcomes to achieve emotional QOL. - Engaging mindfulness practices as a way of reducing distress about future outcomes. - Accepting not being completely well or “cured”. - Hope vested in transcendence and/or acceptance: “letting go” of outcomes, meaning making, trusting doctors/medical science, deferring to God, accepting uncertainty, accepting something less than being completely well or disease free.

**Table 2 cancers-16-00342-t002:** An outcome orientation for qualities of hope based on uncertainty and control.

**Control**	**Low**	**Q3. Hope vested in** **Transcendence** Outcomes uncertain, vested in others, external forces medical science, God, the Universe Letting go, adaptive, disengagement, trust	**Q4. Hope vested in** **Acceptance** Outcomes certain, controlled by others, external forces Acceptance, benefit finding, meaning making, disengagement, denial, despair
	**High**	**Q2. Hope vested in** **Endurance** Outcomes uncertain but vested in persistent, personal effort Endurance, resilience, exhaustion	**Q1. Hope vested in** **Self-Efficacy** Outcomes vested in agency, prior successes in similar situations Self-efficacy, confidence
		**Uncertainty**	
		**High**	**Low**

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
