# Peer review of "Integration of Psychosocial Theory into Palliative Care: Implications for Care Planning and Early Palliative Care"

_cancers, 2024, doi:10.3390/cancers16020342_

Round 1

Reviewer 1 Report

Comments and Suggestions for Authors

Comments

Line 42: define NCI

Table 2 – Midlevel Agency – rephrase “reconciling what appears to be a “disconnect” between agency, which involves patients’ assuming some control their care and the uncertainty of longer-term outcomes”

Table 2 – Advanced Phase – “Relinquishing control of outcomes may be based in religion (placing outcomes in “God’s hands)” – is spirituality or is it religion that the Authors are advocating in this proposal? Should people struggling with end-of-life decision processes be pushed into any ”God” and their hands? Authors might need to reconsider and apply a broader/more inclusive terminology and approach. The patient’s own belief system and/or expectations should guide such broad guideline attempts. Also, syntax revision is needed in this section.

Table 2 – last lines of 3rd section need syntax revision. Also, please revise “God’s hands” narrative to be more inclusive.

Line 112: “as targeted and immunotherapies” – please explain

Line 156: please use specific references from Kant’s and Mill’s work for this comment

Line 251: again – why opposed to religious? Why is religious beliefs the norm in this narrative? Research referenced here states that there are “letting go” techniques in secular and spiritual form – why, then, do the Authors start and compare things to religiosity first?

Line 273 – typo?

Line 290- why not patient-based? Why the bipolar adhesion to religion (first) – or – other?

Line 316 – missing a “

Line 328 – reference for mentioned research

Comments on the Quality of English Language

Minor Syntax revision

Author Response

Responses to Reviewer 1’s comments are in italics after each point.

Thank you for the very detailed reading of the manuscript. The changes based on your comments have added clarity to the narrative. Revisions and emendations based on your comments are highlighted in the revised manuscript in gray.

Line 42: define NCI

This abbreviation has been expanded to show the entire name of the organization.

Table 2 – Midlevel Agency – rephrase “reconciling what appears to be a “disconnect” between agency, which involves patients’ assuming some control their care and the uncertainty of longer-term outcomes”

This phrase in what is now Table 1 was reworded and is hopefully more simply and clearly presented.

Table 2 – Advanced Phase – “Relinquishing control of outcomes may be based in religion (placing outcomes in “God’s hands)” – is spirituality or is it religion that the Authors are advocating in this proposal? Should people struggling with end-of-life decision processes be pushed into any ”God” and their hands? Authors might need to reconsider and apply a broader/more inclusive terminology and approach. The patient’s own belief system and/or expectations should guide such broad guideline attempts. Also, syntax revision is needed in this section.

Thank you for this comment. It was not our intention to focus excessive attention on religious perspectives on relinquishing personal control compared to other approaches. We have attempted to present a more balanced approach in Table 1 as well as in the text where “letting go” is discussed.  We now present religious, spiritual, and secular perspectives on relinquishing control where relinquishing control was mentioned.  

Table 2 – last lines of 3rd section need syntax revision. Also, please revise “God’s hands” narrative to be more inclusive.

As noted above we revised this narrative to be more inclusive.

Line 112: “as targeted and immunotherapies” – please explain

We did include parenthetical explanations of targeted and immunotherapies.

Line 156: please use specific references from Kant’s and Mill’s work for this comment

We included a reference (Kerner, 1990) from which our comment about autonomy was derived.

Line 251: again – why opposed to religious? Why is religious beliefs the norm in this narrative? Research referenced here states that there are “letting go” techniques in secular and spiritual form – why, then, do the Authors start and compare things to religiosity first?

Again, thank you for pointing out the need to be more balanced in the presentation of relinquishing control. We changed the text to reflect a more inclusive, balanced picture of “letting go.”

Line 273 – typo?

Sorry, we could not find the typo in this line.

Line 290- why not patient-based? Why the bipolar adhesion to religion (first) – or – other?

In our reading of the literature, need-based is tantamount to patient-based and is used in the context of differentiating that perspective from prognosis-based.

And, again, thanks for recommending a balance in our presentation. We edited this section to reflect that balance.

Line 316 – missing a “

Thanks for noticing this – corrected!

Line 328 – reference for mentioned research

Reference added.

Reviewer 2 Report

Comments and Suggestions for Authors

The MS submitted by Merluzzi et al titled  " a Psychosocial Component Model of Palliative Care: Implications for Early Palliative Care and Interventions to optimize quality of life" a well written piece for the community. However, I ahve few suggestions to Authors.

1) This type of article improved further by adding a graphical/word diagram describing the main theme of the MS.

2) Rather than writing Simple summary of the MS. Authors should write the highlights in bullet points in brief & consize. 

3) Correct the table numbering in the main text.

4) Try to add few more recent refrences from past few years . 

Author Response

Responses to Reviewer 2’s comments are in italics after each point.

Thank you for your comments. Revisions and emendations based on your comments are highlighted in the revised manuscript in light blue.

The MS submitted by Merluzzi et al titled "a Psychosocial Component Model of Palliative Care: Implications for Early Palliative Care and Interventions to optimize quality of life" a well written piece for the community. However, I have few suggestions to Authors.

1) This type of article improved further by adding a graphical/word diagram describing the main theme of the MS.

We have now included a figure(Figure 2) with the overall model presented. As the reviewer will note this manuscript adds a third dimension (Theoretical Perspectives) to standard palliative care.

2) Rather than writing Simple summary of the MS. Authors should write the highlights in bullet points in brief & concise. 

Because the simple summary is required, we incorporated your recommendation into the simple summary by enumerating the three theoretical perspectives that are the focus of the paper. Our hope is that this clarifies the thrust of the paper, which is that we are elaborating upon and integrating current psychosocial theory into the standard model of palliative care.

3) Correct the table numbering in the main text.

Thanks for noticing this. Done!

4) Try to add few more recent references from past few years.

We did add some references that are more timely.

Reviewer 3 Report

Comments and Suggestions for Authors

Authors present a review article on a new Psychosocial Component Model of Palliative Care  that has three components with implications for interventions to optimize quality of life: Control Theory, Optimal Matching Theory, Hope Theory. What this manuscript lacks is a clear idea of what it wants to present in opposition to - what? standards of care? current palliative medicine? A historical perspective is missing, which should clearly point out the development of the psychosocial component and its limitations and pros compared to the standard of care. A practical illustrative example would explain a lot more about the concept than the whole lot of text, which should be shortened. 

Comments on the Quality of English Language

Acceptable. 

Author Response

Responses to Reviewer 3’s comments.

Authors present a review article on a new Psychosocial Component Model of Palliative Care that has three components with implications for interventions to optimize quality of life: Control Theory, Optimal Matching Theory, Hope Theory. What this manuscript lacks is a clear idea of what it wants to present in opposition to - what? standards of care? current palliative medicine? A historical perspective is missing, which should clearly point out the development of the psychosocial component and its limitations and pros compared to the standard of care. A practical illustrative example would explain a lot more about the concept than the whole lot of text, which should be shortened. 

Revisions and emendations based on your comments are highlighted in the revised manuscript in yellow.

Thank you for drawing attention to the need to clarify how the presentation of psychosocial theory integrates with palliative care. We are not proposing a framework that is in opposition to current palliative care. On the contrary, we are proposing to integrate a conceptual framework that has the potential to enhance the implementation of palliative care and that is applicable throughout the entire trajectory of palliative care from early palliative care to referral to hospice care or release from palliative care. However, our goal is not to relegate the impact of this theoretical enhancement to just psychological, social and spiritual aspects of palliative care. Our view is more wholistic in that the integration of psychosocial theory may, indeed, enhance the psychological, social and spiritual aspects, but also is useful in the physical health realm of palliative care (See Figure 1, which has been added to the manuscript). Our reading of the palliative care literature is that the four dimensions (physical, psychological, social and spiritual) of palliative care are often viewed as separate “services” or entities (with professional roles relegated to separate occupations) with minimal overlap. However, they are all embodied in the patient and carer in spite of the separation conceptually. Our goal was to use psychosocial theory to conceptually enhance palliative care across these four areas and, in doing so, perhaps enhance the provision palliative care. For example, in the standard practice of palliative care, there may be some medical regimens that are taught to a carer or patient to reduce problematic symptoms and enhance the physical well-being of the patient– such as using a Pleurx drain kit to reduce fluid in the pleural cavity. This regimen can be cast in the context of the theoretical perspectives as well as the dimensions of palliative care (See Figure 1). Thus, besides the physical comfort provided (less pain, easier breathing), the process of successfully draining fluid has a psychological component (the patient’s or carer’s agency for mastering the skills needed to perform the procedure, the need to match technical support with level of agency of the carer and patient, dealing with the stress and anxiety of performing a detailed serile procedure, worry and uncertainty about its effectiveness beyond the immediate relief, hope that it will be effective each time ), a social component (the interaction, collaboration and provision of support between the care staff, patient and carer in managing the many intricate steps in the successful execution of the procedure, the bi-directional matching of the provision of support to endure the procedure on a daily basis, hope that one can maintain the support needed to perform the procedure and endure the discomfort ) and a spiritual component (affirmation, connectedness, social presence, mutuality, coping with uncertainty of outcomes, faith, trust)  – cf., Kellehear [2000] Palliative Medicine “Spirituality and palliative care: A model of needs”). We included a version of this integration of the theoretical components with the dimensions of palliative care (physical, psychological, social, and spiritual).

Throughout the manuscript we have attempted to clarify the goals of the presentation according to the rationale described above in response to your insightful comments. Those emended passages and sections are highlighted. In addition, your comment also led to a change in the title to reflect the goal of integrating a psychosocial theoretical framework to enhance palliative care.  

Round 2

Reviewer 3 Report

Comments and Suggestions for Authors

Authors have sufficiently responded to reviewer remarks. 

Comments on the Quality of English Language

Acceptable.